# Targeted High-Resolution Taxonomic Identification of *Bifidobacterium longum* subsp. *infantis* Using Human Milk Oligosaccharide Metabolizing Genes

**DOI:** 10.3390/nu13082833

**Published:** 2021-08-18

**Authors:** Lauren Tso, Kevin S. Bonham, Alyssa Fishbein, Sophie Rowland, Vanja Klepac-Ceraj

**Affiliations:** Department of Biological Sciences, Wellesley College, Wellesley, MA 02481, USA; ltso@wellesley.edu (L.T.); kbonham@wellesley.edu (K.S.B.); afishbein@wellesley.edu (A.F.); srowland@wellesley.edu (S.R.)

**Keywords:** human milk oligosaccharide metabolizing genes, *Bifidobacterium infantis*, metagenomics, comparative genomics, child development, pediatric cohort, metabolic potential

## Abstract

*Bifidobacterium longum* subsp. *infantis* (*B. infantis*) is one of a few microorganisms capable of metabolizing human breast milk and is a pioneer colonizer in the guts of breastfed infants. One current challenge is differentiating *B. infantis* from its close relatives, *B. longum* and *B. suis*. All three organisms are classified in the same species group but only *B. infantis* can metabolize human milk oligosaccharides (HMOs). We compared HMO-metabolizing genes across different *Bifidobacterium* genomes and developed *B. infantis*-specific primers to determine if the genes alone or the primers can be used to quickly characterize *B. infantis*. We showed that *B. infantis* is uniquely identified by the presence of five HMO-metabolizing gene clusters, tested for its prevalence in infant gut metagenomes, and validated the results using the *B. infantis*-specific primers. We observed that only 15 of 203 (7.4%) children under 2 years old from a cohort of US children harbored *B. infantis*. These results highlight the importance of developing and improving approaches to identify *B. infantis*. A more accurate characterization may provide insights into regional differences of *B. infantis* prevalence in infant gut microbiota.

## 1. Introduction

Human microbiota has coevolved with humans for tens of thousands of years [1]. This coexistence requires a tight partnership between the host and its resident microorganisms. For example, the abundant oligosaccharides found in human milk cannot be digested by the infant who feeds on them. Instead, these are used to support the growth of the pioneer gut colonizers, who then, in turn, support the development of the infant [2,3]. In particular, taxa in the *Bifidobacterium* genus often have genes that allow them to metabolize the complex carbohydrates in their hosts [4,5,6]. The subspecies *Bifidobacterium longum* subsp. *infantis* (*B. infantis*), for example, can import and degrade a variety of complex human milk oligosaccharides (HMOs) [7]. Organisms belonging to the same *longum* subgroup, *B. longum* subsp. *suis* (*B. suis*) and *B. longum* subsp. *longum* (*B. longum*), are unable to metabolize HMOs.

The functional potential of *B. infantis* to metabolize HMOs lies in five genomic loci dedicated to the import of HMOs and their processing [8]. The 79 genes within these five clusters reveal substantial glycolytic potential [8], allowing for the transmembrane import of milk oligosaccharides into the cytosol and later cleaving into monosaccharides [9]. The proximity of these genes indicates a role in the regulation of a common metabolic pathway that has coevolved with milk components to outcompete other microbes in the infant gut [8]. In the developing infant gut, where milk is the primary source of nutrients, this ability provides a selective advantage [9]. Given that other taxa of interest may also have differential functional potential, high-resolution identification of all species in the community is desirable. One of the most widely used culture-independent methods to taxonomically profile gut microbiomes, 16S rRNA amplicon sequencing, cannot differentiate *B. infantis* from its closest relatives [10,11,12,13]. Whole metagenome (“shotgun”) sequencing can provide additional taxonomic resolution, but existing tools for assembly-free taxonomic assignment also commonly fail to differentiate subspecies [14,15] *infantis* from other *B. longum* subspecies. It remains unclear whether this HMO metabolic potential is confined to a single lineage or is shared across closely related organisms (e.g., through horizontal gene transfer). The origins of these gene cassettes also remain obscure, though other species such as species belonging to *Bifidobacterium*, *Lactobacillus*, and *Bacteroides* genera have HMO use potential [9,16,17,18,19,20,21]. Understanding the distribution of these genes in microbial species may enable using metagenomic sequencing for the interrogation of *B. infantis* colonization, succession, and distribution in the infant microbiome in new stool metagenomes as well as those that are already available [9]. In this study, we tested the *in vitro* and *in silico* use of these 79 HMO-metabolizing genes and determined that they alone can identify *B. infantis* in a stool sample and assess its prevalence in the population. We then applied this characterization of *B. infantis* to search for its presence in a US pediatric cohort with shotgun metagenomic sequencing. We validated these results by designing and benchmarking subspecies-specific PCR primers targeting a subset of these HMO-metabolizing genes.

## 2. Materials and Methods

### 2.1. Bacterial Genomes

The genome sequences of 387 *Bifidobacterium* species and subspecies were downloaded in FASTA format from the National Center for Biotechnology Information (NCBI) database. Genomes included 4 *Bifidobacterium breve* and 383 strains belonging to *Bifidobacterium longum* group, spanning all three subspecies (43 *B. infantis*, 167 *B. longum*, 3 *B. suis*). In addition, 170 of the genomes were unspecified *B. longum* group strains and were also included in the analysis (Appendix A).

### 2.2. Calculation of ANI Values

All pairwise, whole-genome sequence comparisons were performed using average nucleotide identity (ANI). ANI is an in silico substitute for DNA-DNA hybridization (DDH) and thus is useful for delineating species boundaries [22]. The Python module pyani (v 0.2.7) was used to calculate ANI values [23]. Specifically, the genomic sequence from one of the genomes in a pair was aligned against the other using the matching algorithm MUMmer (v 4.0.0) through the ANIm command [24]. The calculated ANI value was the average percentage nucleotide identity for all matching regions between a pair of genomes. A percentage identity table returned from this analysis was then converted to a distance matrix using R 3.6.0 (Appendix A). The neighbor-joining ANI tree consisting of 387 *Bifidobacterium* genomes was built using the default parameters with the phytools (v 0.6-99) package. Each tree was rooted by the four *B. breve* genomes included as an outgroup and bootstrapped with values assigned to internal edges over 100 iterations.

### 2.3. Genomic Analysis of HMO-Metabolizing Clusters

Specificity of the HMO-metabolizing gene cluster was determined in published genomes from NCBI and sequenced metagenomes from fecal samples. The presence of HMO-metabolizing clusters in each of the *Bifidobacterium* genomes from NCBI was determined by BLASTN hits and filtered for 95% identity. Heatmaps were colored by the presence of at least one BLAST hit to a gene. The presence of HMO-metabolizing clusters in the non-redundant database from NCBI was determined by BLASTP hits and filtered for 80% identity [25].

Evolutionary pressures of the 78 protein-coding HMO-metabolizing genes were assessed by differences in the substitution rates at nonsynonymous (dN) and synonymous sites (dS) using MEGA X (v 10.1.8) [26,27]. A significant difference in the relative abundance of synonymous and nonsynonymous substitutions implies non-neutral selection, with a higher ratio of nonsynonymous substitutions indicating positive selection and a lower ratio indicating purifying selection [28].

### 2.4. Subjects and Samples

Samples (n=1092) used in this study were collected from the RESONANCE Cohort (Providence, RI), an accelerated-longitudinal study of healthy infants and children. Children in the cohort (n=583) were between 33 days and 15 years old (mean=4.47 years), with 144 samples collected from children less than 1 year old, and 205 from children under 2 years old (Appendix A). Approximately 70% of subjects were born vaginally, and approximately 80% were fed at least some breastmilk. All procedures and experiments in the study were approved by and followed the guidelines of the local review board. Written consent was obtained from all parents or guardians of participants. Fecal samples were collected by patients in OMR-200 tubes (OMNIgene GUT, DNA Genotek, Ottawa, ON, Canada), brought to the RI lab and frozen within 24 h at −80 ${}^{\circ }$C in the RI lab.

### 2.5. Sample Processing and Sequencing

All samples were processed at Wellesley College (Wellesley, MA, USA). Nucleic acids were extracted from stool samples using the RNeasy PowerMicrobiome kit automated on the QIAcube (Qiagen, Germantown, MD, USA), excluding the DNA degradation steps. Extracted DNA was sequenced using shotgun metagenomic sequencing at the Integrated Microbiome Resource (IMR, Dalhousie University, Halifax, NS, Canada). A pooled library (max 96 samples per run) was prepared using the Illumina Nextera Flex Kit for MiSeq and NextSeq (a PCR-based library preparation procedure) from 1 ng of each sample where samples were enzymatically sheared and tagged with adaptors, PCR amplified while adding barcodes, purified using columns or beads, and normalized using Illumina beads or manually. Samples were then pooled onto a plate and sequenced on the Illumina NextSeq 550 platform using 150 + 150 bp paired-end “high output” chemistry, generating 400 million raw reads and 120 Gb of sequence. The list of samples used in this study is listed in Appendix A. All metagenomic data were deposited in SRA in BioProject PRJNA695570.

### 2.6. Identification of B. infantis in Metagenomic Samples by the Presence of HMO-Metabolizing Clusters

The presence of *B. infantis* in each of the 1092 sample metagenomes included in our study was determined by that of the majority (75%) of the 79 HMO-metabolizing genes in the gene cluster. Metagenomic samples were analyzed using the bioBakery [29] family of tools with default parameters. Briefly, KneadData (v 0.7.1) was used to trim and filter raw sequence reads and to separate human reads from bacterial sequences. Samples that passed quality control were taxonomically profiled to the species level using MetaPhlAn (v 3.0.1). Paired-end reads of the 150 bp nucleotide sequences acted as inputs against a reference database including the 79 HMO-metabolizing genes using the alignment tool Bowtie2 (v 2.4.2) [30]. Bowtie2 was used in these searches, and not in any others, for its specific ability in aligning short reads to relatively long genomes. The presence of a gene was noted if at least one mapped alignment was found to that gene.

### 2.7. Primer Design and Testing

Previously, published primers designed to target *B. infantis* were either too specific or did not target *B. infantis* or were too broad and targeted other members of the *B. longum* group (Table 1; [2,31,32]). We decided to test the existing primers *in silico* and to develop new primer sets. To find candidate regions, we analyzed 79 genes associated with the five HMO-metabolizing clusters and selected genes that were present in all *B. infantis* and that had gene sections highly conserved across the different strains.

We chose four HMO-metabolizing genes that aligned >97% to each *B. infantis* but not to any other bacteria and the included: exo-α-sialidase gene (locus tag ’Blon2348’), glycoside-hydrolase gene (locus tag ’Blon2355’), haloacid dehalogenase-like hydrolase domain-containing protein gene (locus tag ’Blon2356’) and β-lactamase domain protein gene (locus tag ’Blon2358’). To test the target specificity of the published primers and to design the new primers, 48 *B. longum* group genomes and two *B. suis* genomes were imported into Geneious (v R11.1.5), see Appendix A. We aligned the genes from the selected genomes using MUSCLE (v 3.6) [35] alignment in Geneious and used them to design the primer pairs via Primer3 (v 2.3.7) [36]. Each primer pair was checked using NCBI primer-BLAST to ensure that it only matches *B. infantis* sequences [37]. All primers targeting HMO-metabolizing genes were checked against 387 genomes (Appendix A) to assess their coverage (Table 1). Primers were obtained from Integrated DNA Technologies, Inc., (Coralville, IA, USA).

To ensure that the primers were adequate for targeted detection of *B. infantis* we used genomic DNA from isolated cultures of *B. infantis* (DSM 20088), *B. longum* (DSM 20219), *B. breve* (DSM 20213) and *E. coli* K12. The cultures were purchased from the German Collection of Microorganisms and Cell Cultures GmbH (DSMZ, Leibniz, Germany). The *Bifidobacterium* strains were incubated anaerobically at 37 ${}^{\circ }$C for 36 h. The cells of each culture were centrifuged at 13,000 rpm for 1.5 min and washed in anaerobic PBS. Genomic DNA was then extracted using DNeasy PowerSoil Pro Kit (Qiagen, Germantown, MD, USA) following manufacturer’s instructions. DNA concentrations were determined using an absorbance ratio of 260/280 nm.

### 2.8. Development of PCR Conditions

PCR amplifications were performed using OneTaq Quick-Load 2X Master Mix (New England BioLabs, Ipswich, MA, USA), 200 nM of each primer, and 10 ng of template DNA (25 µL total volume) following manufacturer’s instructions. After the initial denaturation step at 94 ${}^{\circ }$C for 30 s, there were 30 cycles of 94 ${}^{\circ }$C for 30 s, the respective annealing temperature for 60 s, and 72 ${}^{\circ }$C for 60 s, and then finally 68 ${}^{\circ }$C for 5 min. The positive control was gDNA of isolated cultures of *B. infantis* (DSM 20088) or of *B. breve* (DSM 20213) and the negative control contained water instead of a template DNA were included with each batch.

### 2.9. Confirmation of HMO-Metabolizing Genes in the Metagenomes by PCR

To detect *B. infantis* by PCR in fecal samples, we used the *B. infantis* exo-α-sialidase gene (locus tag ‘Blon2348’) primer pair (Sia-266F and Sia-676R). We also used 16S rRNA gene primers that target the genus *Bifidobacterium* (g-Bif-F and g-Bif-R) or universal 16S rRNA primers (27F and 1492R). All primers were purchased from Integrated DNA Technologies (IDT, Coralville, IA, USA).

## 3. Results

The first step in assessing the suitability of functionally differentiating *B. infantis* from other members of the *B. longum* group was to assess the phylogenetic relationships among members of this group. To accomplish this, we measured the average nucleotide identity (ANI) and shared gene content of all (387) publicly available *B. longum* group genomes (Appendix A). Four *B. breve* genomes (GCF_000158015.1, GCF_000213865.1, GCF_000220135.1, GCF_000226175.1) were included as an outgroup. Genomes in this analysis were stratified by the (sub)species in each pairwise comparison (Figure 1). As expected, genomes of the same species were the most similar, and within *B. longum*, subspecies were all >95% similar to one another (Appendix A). Subspecies clusters (*B. longum*, *B. infantis*, and *B. suis*) were clearly distinct by ANI, though substantially overlap in terms of shared gene content. By both metrics, *B. longum* and *B. infantis* were the most similar of the subspecies (Figure 1). Interestingly, each of these subspecies appears to separate into two groups by ANI. As expected, comparisons between the outgroup *B. breve* genomes and *B. longum* subspecies were the least similar.

Though *B. infantis* is resolvable from other *B. longum* subspecies by ANI, performing ANI calculations on metagenome assemblies is computationally expensive and requires high sequencing depth [38]. We therefore set out to determine whether individual genes might be suitable for distinguishing these subspecies. We explored the possibility that the presence of five genomic loci containing HMO-metabolizing genes could be used as a proxy for subspecies differences (Figure 2). To do so, we used BLAST to quantify the presence of the five subspecies-specific gene clusters within the same 387 *Bifidobacterium* genomes. Genomes previously characterized as *B. infantis* largely corresponded with both ANI-based clustering and with the presence of HMO-metabolizing genes, with 36 of 43 *B. infantis*-labeled genomes (Appendix A), found in a distinct clade (Figure 2, top clade) and containing at least 93% of HMO-metabolizing genes. Crucially, seven out of these 43 *B. infantis*-labeled genomes, including 5 strains (CCUG 52486, CECT 7210, JCM 11660, KCTC 5934, 157F), did not contain most HMO-metabolizing genes, but also did not cluster with other *B. infantis* genomes based on ANI, suggesting that they may have been misclassified. These results persist when HMO-metabolizing genes were excluded from ANI calculations (Figure 2, Appendix A), precluding the possibility that the ANI similarities are solely attributable to the presence or absence of those genes. Conversely, five unspecified *B. longum* genomes (GCF_900157085.1, GCF_900157065.1, GCF_900157185.1, GCF_900157075.1, GCF_902381625.1) clustered tightly with those from *B. infantis* strains and should, perhaps, be more specifically labeled as this particular subspecies (Figure 2). These genomes, although labeled as *B. longum*, contained HMO-metabolizing genes characteristic for *B. infantis*.

The congruence of ANI and presence of HMO-metabolizing genes suggests that *B. infantis* diverged from other *B. longum* subspecies since the acquisition of these genes. To explore the history of the HMO-metabolizing gene clusters in more detail, we investigated their presence within and beyond the *Bifidobacterium* genus. We used BLAST to search for homologs of the 79 genes from the cluster against all available proteins in the NCBI database. Hits (see methods) for each of the HMO-metabolizing genes came largely from *B. longum* and *B. breve* genomes (Figure 3A, Appendix A). Cluster 1 displayed the highest percentage of hits to genomes from both of these species at 64% and 24% respectively.

Because microbial genomes can be quite labile [39], we set out to assess whether the genes targeted by this approach were likely to remain stable by exploring the evolutionary pressures acting on the HMO use cassettes. To do so, we carried out systematic BLAST searches on each of the 78 protein-coding genes out of the 79 HMO-metabolizing genes. We hypothesized that genes under stabilizing selection would be better genomic markers of a subspecies, since these genes would be less likely to suffer gene loss or rapid mutation that would disrupt primer binding. We compared nucleotide differences in the substitution rates at nonsynonymous (dN) and synonymous sites (dS) using MEGA X [26,27]. A significant difference in the relative abundance of synonymous and nonsynonymous substitutions implies non-neutral selection, with a higher ratio of nonsynonymous substitutions indicating positive selection and a lower ratio indicating purifying selection [28]. Of the 78 protein-coding genes across the five clusters, 61 have undergone stabilizing selection (Appendix A, Figure 3B). Twelve genes underwent positive selection and five did not show evidence of selection (neutral selection) (Figure 3B).

After establishing that the five HMO-metabolizing gene clusters can distinguish *B. infantis* from the rest of the *B. longum* group members, we assessed the suitability of using HMO-metabolizing genes to generate primers specific to *B. infantis*. The development of the new primer sets was motivated by realizing that the existing primers to profile *B. infantis* were either too broad (e.g., 16S rRNA gene [33]) or too narrow (e.g., primers targeting sialidase gene [32]). Forty-six out of 79 HMO-metabolizing genes covered were found to be in at least 85% of *B. infantis* strains. Two of these genes (Blon0218 and Blon0219) were found exclusively in the *B. infantis* clade (Figure 2, top clade) and might serve as potential future targets for primer development. Eleven additional genes (Blon2322, Blon2335, Blon2336, Blon2337, Blon2338, Blon2348, Blon2349, Blon2355, Blon0220, Blon0222, Blon0232) were present in all genomes within the *B. infantis* clade and one additional *B. infantis* genome (GCF_902167615.1) right outside the clade. This genome (GCF_902167615.1) contains a majority of HMO-metabolizing genes found in Cluster 1 (Figure 2) and would be classified as *B. infantis* by our analysis as well if it were limited to this HMO-metabolizing gene cluster.

Out of the 11 genes, we selected four genes (Blon2348, Blon2355, Blon2356 and Blon2358) that aligned well with the clade of 41 genomes we consider to be *B. infantis* (Appendix A). Genes exo-α-sialidase (Blon2348) and glycoside-hydrolase (Blon2355) were present in all genomes within the clade and one additional *B. infantis* genome (GCF_902167615.1) right outside of it. We also selected genes Blon2356 and Blon2358 that were present in 34 (83%) of the genomes within this clade and the additional *B. infantis* genome (GCF_902167615.1) as well. All four genes also had conserved regions among the *B. infantis*, making them good candidates for primer development.

Primer pairs for these four genes were tested *in silico* against the 387 downloaded genomes (Table 1, Appendix A). The primer specificity to the *B. infantis* clade matched clustering of strains based on ANI (Figure 2). The same six *B. infantis*-labeled genomes that failed to cluster with *B. infantis* by ANI provided no hits to our primer pairs (Appendix A), providing additional evidence that these *B. infantis*-named strains may have been misclassified. Similarly, the five unspecified *B. longum* genomes (GCF_900157085.1, GCF_900157065.1, GCF_900157185.1, GCF_900157075.1, GCF_902381625.1) that were clustered within the *B. infantis* clade (Figure 2) perfectly hybridized *in silico* to our primer sets.

To confirm the specificity of the designed primer pairs *in vitro*, PCR assays were performed with three *Bifidobacterium* reference strains, an *E. coli* K12 strain and a negative control. The amplification product was again specific to *B. infantis* (Table 1, the first 4 primer pairs). All four primers aligned 100% to *B. infantis* and not to any other bacteria. These primers were then applied to verify *B. infantis* presence in sequenced fecal samples.

Finally, we used the five HMO-metabolizing gene clusters to distinguish *B. infantis* from the rest of the *B. longum* subspecies in the gut microbiomes of human children using fecal samples in the RESONANCE cohort of young, healthy children [40]. We applied the specificity of this HMO-metabolizing cluster to identify *B. infantis* from metagenomic analysis of 1092 fecal samples. These samples included pregnant mothers, and children ranging from infants to adolescents. Sequences belonging to the Bifidobacterium longum species were also found in the metagenomes of most (158 of 203 or 78%) of the samples from children under two years old analyzed. By searching for the HMO-metabolizing gene clusters in shotgun metagenomic data, we observed that 15 of 203 (7.4%) samples from children under the age of two harbor *B. infantis* (Figure 4). The gut microbiomes of three of the subjects were sampled at later times and no longer contained *B. infantis*, suggesting a decrease in abundance around one year of age. PCR amplification of one HMO-metabolizing gene (the sialidase gene Blon2348) in a subset of samples corroborated the metagenomic evidence (Figure 4). When screened by PCR with *B. infantis*-specific primers, only 13 of 162 (8.0%) samples taken from children under two and 4 of 411 (1.0%) samples taken from mothers were found to have *B. infantis*.

## 4. Discussion

The human gut microbiome, though better characterized than most microbial communities, remains poorly understood. This is especially true in very young children, whose gut microbial communities are seeded at birth, shaped profoundly by the method of their birth and the nature of their diet, and in constant flux. The identification nearly 100 years ago of a component of human breast milk (HMO) that appears to have evolved solely to feed the developing microbiomes of infants hints at the critical importance of microbiome development as part of overall child development [41]. Here, we have shown that by focusing on the functional differences between microbial subspecies, in particular those differences that provide a critical metabolic function such as HMO use, we can increase taxonomic resolution through culture-independent means.

The *Bifidobacterium* genus is incredibly common in children’s guts. In this study, 96% of children have them as confirmed by PCR assay and metagenomic search and these findings are consistent with other early life gut microbiome studies [42,43]. Within the *Bifidobacterium* genus, many species harbor genes that allow them to make use of complex polysaccharides. Yet, there are important functional differences between members of this genus, and even, as we have shown here, between the subspecies *B. longum* and *B. infantis*. This is not uncommon; a recent study showed an average of 13% (and as much as 25%) difference in gene content between common gut microbiome species from different individuals [39]. Recent comparisons of the infant gut microbiome indicate that *B. infantis* is more prevalent among infants from countries with limited resources such as Gambia (76.9%) and Bangladesh, and less prevalent in more developed countries such as Finland (10%), with increasing prevalence in countries such as Estonia (20%) and Russia (23%) [44,45]. Using metagenomic sequencing and a gene function-centered approach allowed us to show that only 7.4% children under 2 years old in a cohort of US children harbored the subspecies *B. infantis*, and we showed that this is synonymous with harboring five HMO-metabolizing gene clusters. These results are on par with a recent study on microbial dysbiosis in American infants in the first six months of life that showed remarkably low prevalence of HMO-metabolizing capacity of 10% that accompanied other indicators of unhealthy microbiomes [46]. The lower prevalence of *B. infantis* in our cohort is likely due to inclusion of children who have transitioned from the diet of human milk to solid foods. Given the apparent decline of *B. infantis* in western populations, and the beneficial associations of *B. infantis* with allergies and autoinflammatory disorders [47,48], other health conditions [49], and healthy brain development [50], it is critical to have accurate classification methods for this microbe to improve our ability to understand the links between HMO use and early child development.

This study addresses some of the existing limitations for studying *B. infantis*. Although amplicon (16S rRNA gene) sequencing is typically only capable of resolving genus-level differences between microbes, both PCR and shotgun metagenomic sequencing have the capacity to interrogate gene-level variation. Yet these two methods have important trade-offs; PCR-based detection is straightforward, cost-effective, and able to detect very rare gene functions, but difficult to scale to large numbers of samples or to identify many gene functions at once. Shotgun metagenomic sequencing can be readily scaled, but is substantially more costly, especially at sequencing depths necessary to detect low-abundance genes. Here, we showed that metagenomic sequencing largely agrees with PCR-based amplification, at least for one HMO-metabolizing gene. A custom primer pair designed to target the HMO-metabolizing sialidase gene, Blon2348, in *B. infantis* was used in PCR in this study. Eleven of the 79 HMO-metabolizing genes were also found in all genomes within the *B. infantis* ANI cluster; a subset of these might be used for future primer development. Further using PCR to investigate the presence of many genes in multiple gene clusters, however, would have been impractical, and the concordance of dozens of genes using metagenomic sequencing lends increased credence to the presence of these genes in the samples identified through this method.

We were able to identify six genomes that were previously identified as *B. infantis* that should in fact be classified as *B. longum*, as well as five genomes that do not have subspecies resolution in Genbank that likely should be classified as *B. infantis*. The high concordance between average nucleotide identity and the presence of all 5 HMO use clusters suggests that acquisition or loss of these genes occurred in a single common ancestor and continued divergence with positive selection due to occupying the HMO use niche. However, this need not always be the case; horizontal gene transfer (HGT) can lead to gene function sharing across species boundaries [51,52]. In human microbiome epidemiology, what likely matters in shaping health outcomes is functional variation, for which taxonomic profiling is an imperfect proxy. We believe that a function-based metagenomic approach is essential for untangling the complexity of the gut microbiome and its relationship to human health.

## Figures and Tables

**Figure 1 nutrients-13-02833-f001:**
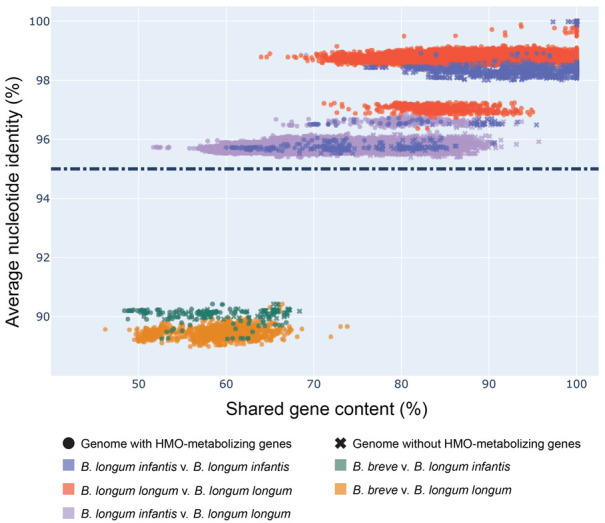
*B. infantis* genomes cluster separately from *B.longum*. Scatter plot of percent shared gene content versus ANI values across pairwise comparisons of *Bifidobacterium* genomes. Shared gene content calculated from BLAST and ANI values calculated by the Python package pyani. Presence of HMO-metabolizing genes determined by the occurrence of at least 60 of 79 (75%) genes in a genome. Dashed line indicates the ANI cutoff score of 95% for genomes belonging to the same species.

**Figure 2 nutrients-13-02833-f002:**
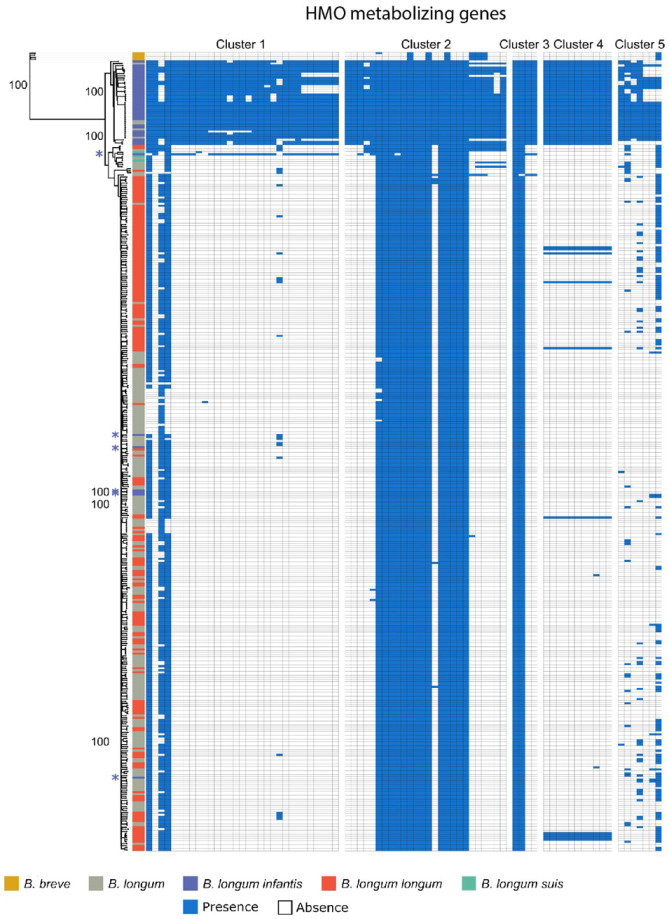
HMO-metabolizing gene clusters mirror phylogenetic separation of *B. infantis*. Rooted neighbor-joining tree of the average nucleotide identity (ANI) values of the same 387 *Bifidobacterium* genomes with HMO-metabolizing genes removed (Appendix A). The tree is paired with a heatmap denoting the presence of HMO-metabolizing genes by BLASTN alignments (>95% identity). Potentially misclassified *B. infantis* genomes noted with a purple asterisk.

**Figure 3 nutrients-13-02833-f003:**
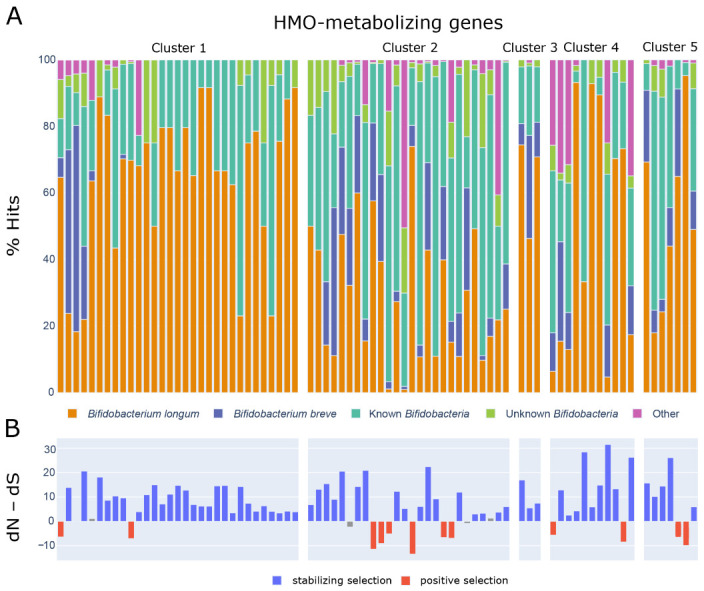
(**A**) Stacked bar plot showing taxonomic assignment of BLASTP hits (NCBI non-redundant database) for each HMO metabolizing gene. Known *Bifidobacterium* include any specified species in the *Bifidobacterium* genus other than *B. longum* or *B. breve*. Unknown *Bifidobacterium* include any unspecified *Bifidobacterium* taxa (*Bifidobacterium*, *Bifidobacterium* spp.). (**B**) Statistics from dN–dS analysis showing the probability and certainty that a given gene within the HMO-metabolizing gene clusters are under stabilizing or positive selection. Colored bars indicate statistical significance (p<0.05). Analyses were conducted using the Nei-Gojobori method in MEGA X. All ambiguous positions were removed for each sequence pair (pairwise deletion option).

**Figure 4 nutrients-13-02833-f004:**
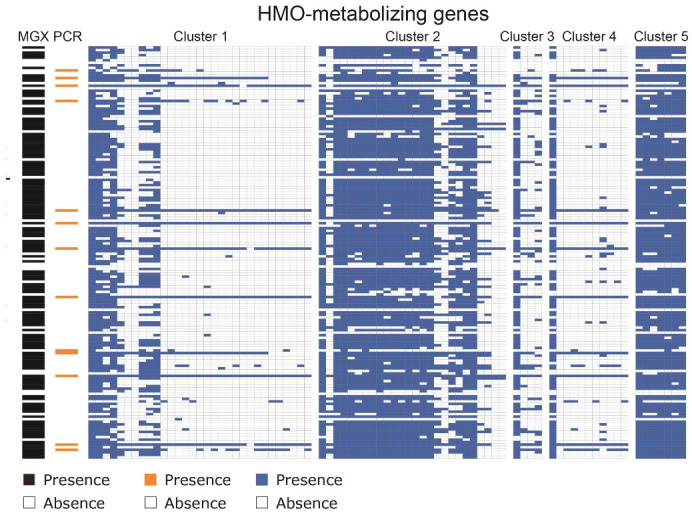
B. infantis presence observed in 15 of 203 (7.4%) samples from children under the age of two via PCR or metagenomics. Results from tests on samples from children under the age of two for the presence of HMO-metabolizing genes either via metagenomics (n=203) or PCR (n=162). The presence of *B. infantis* in metagenomes (n=15) was determined by mapped Bowtie2 alignments to HMO-metabolizing genes. The PCR presence of *B. infantis* (n=13) was determined by PCR using primers designed to target the gene Blon2348 unique to the subspecies, with presence noted in orange and absence in white. Presence of *Bifidobacterium longum* calculated by MetaPhlAn is also noted on the left-hand side in the MGX column in black and absence in white.

**Table 1 nutrients-13-02833-t001:** Primers targeting HMO metabolizing and 16S rRNA genes.

Gene	Target Organism	Name	Sequence (5’-3’)	Annealing Temp (${}^{\circ }$C)	% Coverage ^1^	References	Length (bp) ^2^
exo-α-sialidase (Blon_2348)	*B. longum infantis*	Sia-266F	GACGAGGAGGAATACAGCAG	58	100	This study	410
		Sia-676R	CACGAACAGCGAATCATGGATT				
glycoside-hydrolase (Blon_2355)	*B. longum infantis*	GH-750F	GCGCCATCCTGGTGATGTTATT	59	100	This study	498
		GH-1248R	CTACGTGATCTGGGAGAGTTTC				
haloacid dehalogenase domain protein hydrolase (Blon_2356)	*B. longum infantis*	HH-60F	CCACAATGTCATCGACCATCTG	59	83	This study	474
		HH-534R	CCGAAGTATTCGGATGCCTATG				
glycoside hydrolase (Blon_2358)	*B. longum infantis*	GH-492F	CGATGATGTGCTGGATTCGTTC	58	74	This study	510
		GH-1002R	CTCGACCATTCCAAGATGCTCA				
Sialidase	*B. longum infantis*	Inf 2348-F	ATACAGCAGAACCTTGGCCT	60	50	[32]	280
		Inf 2348-R	GCGATCACATGGACGAGAAC				
Major Facilitator Superfamily	*B. longum infantis*	Blon0915-F	CGTATTGGCTTTGTACGCATTT	50	71	[2]	118
		Blon0915-R	ATCGTGCCGGTGAGATTTAC				
16S rRNA	*B. longum infantis, B. longum longum, B. indicum*	Binf-F	CCATCTCTGGGATCGTCGG	57		[31]	562
		Binf-R	TATCGGGGAGCAAGCGTGA				
16S rRNA	*B. infantis*	HCBin-F	AGGATACGTTCGGCGTC	60		[33]	377
		HCBin-R	CGCAAGATTCCTCTAGCA				
16S rRNA	*Bifidobacterium*	g-Bif-F	CTCCTGGAAACGGGTGG	55		[31]	550
		g-Bif-R	GGTGTTCTTCCCGATATCTACA				
16S rRNA	All bacteria	Uni-27F	AGAGTTTGATCCTGGCTCAG	55		[34]	≈ 1400
		Uni-1492R	RGYTACCTTGTTACGACTT				

^1^ Percent of exact alignment to 42 *B. infantis* strains. ^2^ Amplicon size in base pairs.

## Data Availability

Metagenome data were deposited in NCBI’s Sequence Read Archive (SRA), under BioProject PRJNA695570 and accession numbers SAMN17618458-9435. Code for data analysis, statistics, and figures generated for this manuscript is available on OSF.io [40].

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
