# Peer review of "Targeted High-Resolution Taxonomic Identification of Bifidobacterium longum subsp. infantis Using Human Milk Oligosaccharide Metabolizing Genes"

_nutrients, 2021, doi:10.3390/nu13082833_

Round 1

Reviewer 1 Report

The study is very interesting and very well written but I think that the authors have to explain with more details some issues. My mayor concerns are: 

1)The authors don’t explain which type of population they analyzed and the average of age. It is important to divide children according with the age 

2) There is a lack of information about the subjects of the study: the type of delivery, the type of nutrition (breastmilk versus artificial milk), the gestational age at delivery 

The microbiota is established in early pregnancy and varies depending on maternal nutritional habits and gestational age. The type of child nutrition (breastfeeding or artificial milk) is crucial in the development of microbiota.  (See Navarro-Tapia, E.; Sebastiani, G.; Sailer, S.; Almeida Toledano, L.; Serra-Delgado, M.; García-Algar, Ó.; Andreu-Fernández, V. Probiotic Supplementation during the Perinatal and Infant Period: Effects on gut Dysbiosis and Disease. Nutrients 202012, 2243) 

3)A table with the characteristics of population would be appreciated 

5) In the discussion it would be better to highlight the clinical applicability of this study  

Author Response

Response to Reviewer 1 Comments:

​​The study is very interesting and very well written but I think that the authors have to explain with more details some issues. My mayor concerns are: 

1) The authors don’t explain which type of population they analyzed and the average of age. It is important to divide children according with the age

We thank the reviewer for this comment. See below for the combined response for points 1 through 3.  

2) There is a lack of information about the subjects of the study: the type of delivery, the type of nutrition (breastmilk versus artificial milk), the gestational age at delivery 

Please see below for the combined response for points 1 through 3.  

The microbiota is established in early pregnancy and varies depending on maternal nutritional habits and gestational age. The type of child nutrition (breastfeeding or artificial milk) is crucial in the development of microbiota.  (See Navarro-Tapia, E.; Sebastiani, G.; Sailer, S.; Almeida Toledano, L.; Serra-Delgado, M.; García-Algar, Ó.; Andreu-Fernández, V. Probiotic Supplementation during the Perinatal and Infant Period: Effects on gut Dysbiosis and Disease. Nutrients 2020, 12, 2243) 

We thank the reviewer for bringing this paper to our attention. We’ve cited it in the section added to the discussion about clinical relevance (see response to point 5 below)

3) A table with the characteristics of population would be appreciated 

We thank the reviewer for the suggestion to clarify the composition of the cohort. We have described this cohort in more detail elsewhere (e.g., https://doi.org/10.1101/2020.02.13.944181), but agree that the relevant details are important to include here as well. We have added Supplementary Table 3 with more specific demographic information, and included the following in section 2.4 (ln XXX-YYY):

“Children in the cohort (n=583) were between 33 days and 15 years old (mean=4.47 years), with 144 samples collected from children less than 1 year old, and 205 from children under 2 years old (Supplementary Table 3).  Approximately 70% of subjects were born vaginally, and approximately 80% were fed at least some breastmilk.”

5) In the discussion it would be better to highlight the clinical applicability of this study  

We also appreciate the opportunity to expand our discussion to include additional information about the clinical applicability of our findings. To this end, we have added the following to the discussion:

“Given the apparent decline of B. infantis in western populations, and the beneficial associations of B. infantis with allergies and autoinflammatory disorders [46,47],  other health conditions [48],  and healthy brain development [49], it is critical to have accurate classification methods for this microbe to improve our ability to understand the links between HMO utilization and early child development.”

Reviewer 2 Report

The paper by Tso et al. focuses on the identification of infants subsp.  specific couple of primers, useful to sequence clusters of oligosaccharide metabolizing genes. This is very helpful in avoiding erroneous taxonomic assignments, which generally in 16S metagenetics stop at the genus level. The workflow they followed sounds good and genomics/bioinformatics analyses were properly performed.

Please improve and check supplementary material legends and headers. Add header descriptions, where they are not present. 

As an example, supplementary table 4 reports supplementary fig.3

Some others lack the header. 

Author Response

Response to Reviewer 2 Comments

The paper by Tso et al. focuses on the identification of infants subsp. specific couple of primers, useful to sequence clusters of oligosaccharide metabolizing genes. This is very helpful in avoiding erroneous taxonomic assignments, which generally in 16S metagenetics stop at the genus level. The workflow they followed sounds good and genomics/bioinformatics analyses were properly performed.

We appreciate the positive feedback on the usefulness of our study and the appropriateness of our genomic analysis.

Please improve and check supplementary material legends and headers. Add header descriptions, where they are not present. 

As an example, supplementary table 4 reports supplementary fig.3

Some others lack the header. 

We thank the reviewer for noticing the consistency of our references and the lack of headers in some of our display items, especially in the supplement. We have therefore updated the text and reorganized our tables as such:

Table 1: Primers targeting HMO metabolizing genes and 16S rRNA.​

Supplementary Table 1. Bifidobacterium genomes used in this study.

Supplementary Table 2. Average nucleotide identity between two genomes

Supplementary Table 3. Average nucleotide identity between two genomes with HMO-metabolizing genes removed.

Supplementary Table 4. The composition of the cohort used in this study.

Supplementary Table 5. Fecal sample IDs used in this study.

Supplementary Table 6. Composition of BLASTP alignments of HMO-metabolizing genes to taxa contained in NCBI’s non-redundant database.

Supplementary Table 7. Statistics from dN - dS analysis of HMO-metabolizing genes.